A review of coral bleaching specimen collection, preservation, and laboratory processing methods

McLachlan Rowan H. mclachlan.8@osu.edu 1
Dobson Kerri L. 1
Schmeltzer Emily R. 2
Vega Thurber Rebecca 2
Grottoli Andréa G. grottoli.1@osu.edu 1
1 School of Earth Sciences, Ohio State University , Columbus , OH , United States of America
2 Department of Microbiology, Oregon State University , Corvallis , OR , United States of America
Levy Oren
Electronic publication date: 2021 Jul 8
Publication date: 2021
Volume: 9
Electronic Location ID: e11763
Received 2021 Apr 21; Accepted 2021 Jun 21
Copyright: ©2021 McLachlan et al.
Copyright year: 2021
Copyright holder: McLachlan et al.
License: This is an open access article distributed under the terms of the Creative Commons Attribution License, which permits unrestricted use, distribution, reproduction and adaptation in any medium and for any purpose provided that it is properly attributed. For attribution, the original author(s), title, publication source (PeerJ) and either DOI or URL of the article must be cited.
License URL: https://creativecommons.org/licenses/by/4.0/

Keywords: Coral, Bleaching, Surveys, Experiments, Methods, Specimens, Pretreatment, Handling, Sampling, Samples

Funding: National Science Foundation Division of Biological Oceanography OCE#1838667 OCE#1923836 Major funding for this research came from a National Science Foundation Division of Biological Oceanography grant OCE#1838667 to Andréa G. Grottoli and OCE#1923836 to Rebecca Vega Thurber. The funders had no role in study design, data collection and analysis, decision to publish, or preparation of the manuscript.

==============================
Under current climate warming predictions, the future of coral reefs is dire. With projected coral reef decline, it is likely that coral specimens for bleaching research will increasingly become a more limited resource in the future. By adopting a holistic approach through increased collaborations, coral bleaching scientists can maximize a specimen’s investigative yield, thus reducing the need to remove more coral material from the reef. Yet to expand a specimen’s utility for additional analytic methods, information on how corals are collected is essential as many methods are variably sensitive to upstream handling and processing. In an effort to identify common practices for coral collection, sacrifice, preservation, and processing in coral bleaching research, we surveyed the literature from the last 6.5 years and created and analyzed the resulting dataset of 171 publications. Since January 2014, at least 21,890 coral specimens were collected for bleaching surveys or bleaching experiments. These specimens spanned 122 species of scleractinian corals where the most frequently sampled were Acropora millepora, Pocillopora damicornis, and Stylophora pistillata. Almost 90% of studies removed fragments from the reef, 6% collected skeletal cores, and 3% collected mucus specimens. The most common methods for sacrificing specimens were snap freezing with liquid nitrogen, chemical preservation (e.g., with ethanol or nucleic acid stabilizing buffer), or airbrushing live fragments. We also characterized 37 distinct methodological pathways from collection to processing of specimens in preparation for a variety of physiological, -omic, microscopy, and imaging analyses. Interestingly, almost half of all studies used only one of six different pathways. These similarities in collection, preservation, and processing methods illustrate that archived coral specimens could be readily shared among researchers for additional analyses. In addition, our review provides a reference for future researchers who are considering which methodological pathway to select to maximize the utility of coral bleaching specimens that they collect.

Introduction

Tropical coral reefs harbor an astounding number of associated species and annually provide a wealth of ecosystem and ecological services on a global scale (Moberg & Folke, 1999; Costanza et al., 2014). Their protection in the face of a warming climate remains of the utmost importance. Coastal communities worldwide have witnessed frequent and devastating coral bleaching events over the past decade with no foreseeable reprieve. It is estimated that thousands of square kilometers of coral reefs have bleached and then died from heat stress events in just the last few decades (Hughes et al., 2017b; Hughes et al., 2018a), and this trend will likely increase over time as sea surface temperatures continue to rise (e.g., Sammarco, Winter & Stewart, 2006; Lima & Wethey, 2012; Van Hooidonk et al., 2016; Frölicher, Fischer & Gruber, 2018).

Bleaching occurs when corals undergo severe stress events causing them to expel their endosymbiotic algae. This symbiont loss further limits coral’s ability to recover from heat-stress events and often leads to higher prevalence of disease and mortality (e.g., Jones, 2008; Hughes et al., 2018b; Muller, Bartels & Baums, 2018). Coupled with the increase in severity of tropical storms, ocean acidification from anthropogenic carbon emissions, overfishing, and pollution from human activities, coral reefs are subject to an increasingly hostile environment (Hoegh-Guldberg, 1999; Hoegh-Guldberg, 2014; Frieler et al., 2013; Hughes et al., 2017a). With projected coral reef decline, it is likely that coral specimens (Box 1) for bleaching research will become a more limited resource. Coral bleaching researchers are increasingly adopting collaborative research approaches, thus reducing the need to remove more coral material from the reef. Yet in order to expand a specimen’s utility for additional analytic methods, information on how corals are collected and preserved is essential as many analytical methods are sensitive to upstream handling and processing.

Investigations into coral bleaching and resilience generally follow two main approaches: surveys of corals in situ and manipulative experiments. While both approaches differ substantially in the types of scientific questions they can address, they share several commonalities including: (1) sampling design, (2) specimen collection, (3) specimen preservation, (4) specimen processing, and (5) downstream analysis (see Fig. 1; Box 1). The methods used during steps 2 through 4 are important as they influence the final state and condition of the coral specimen to be analyzed. For example, consider two hypothetical study collection pathways: (1) coral fragments (Box 1) are removed from the reef using sterile tools, immediately snap frozen in liquid nitrogen, stored individually in sterile containers, and maintained at −80 °C at all times during transport to the laboratory, and (2) coral fragments are removed from the reef and immediately immersed in a chemical fixative such as formaldehyde and stored at room temperature during transport to the laboratory. The methods used during each of these pathways are different, but reflect steps which are necessary for different downstream analyses. With pathway 1, coral specimens were handled such that they are suitable for many physiological, isotopic, microbial, genetic, and/or transcriptomic analyses. Conversely, under pathway 2, specimens are collected in a way conducive to histological analysis and/or various microscopy and imaging techniques such as transmission electron microscopy (TEM) or nanoscale secondary ion mass spectrometry (nanoSIMS).

Figure 1 Flowchart illustrating the type of methodological information collected from coral bleaching surveys and bleaching experiments.

This includes (1) sampling design, (2) specimen collection, (3) specimen preservation, (4) specimen processing, and (5) downstream analyses. Images hand-drawn by Rowan McLachlan.

Overall, these scenarios highlight that, while there is unlikely a universal approach to these collection and processing methods, identification of critical factors in these techniques that expand specimens utility could provide a decision framework for researchers when planning future coral collection, preservation, and processing. Any resulting modifications could maximize collaboration potential and the number of scientific deliverables from fewer specimens. In other words, addressing and documenting some critical aspects of specimen collection, preservation, processing, and storage has the potential to foster increased collaboration through specimen sharing, increased number of analyses per specimen leading to a greater understanding of coral responses to bleaching, and ultimately could lead to both a reduction in cost, effort, and coral sacrifice in coral bleaching research.

Using a dataset generated from a survey of 171 peer-reviewed journal articles published between January 2014 and August 2020, we complied a review of the common practices for sampling, preserving, and processing coral specimens across multiple coral bleaching research disciplines. The goals of this review were to: (1) document the diversity and abundance of corals sampled during bleaching surveys and experiments over the last 6.5 years, (2) catalogue the extent of methodological variation in coral bleaching specimen collection, preservation, and laboratory processing, and (3) identify the downstream analyses conducted following each of these methodological pathways. By summarizing this data, we aim to create a dataset which can be used as a tool for researchers to identify potential specimens available for collaboration, as well as provide context for future bleaching researchers who are considering which methodological pathway to select for a given experiment in order to maximize the downstream utility and yield of newly collected specimens.

BOX 1 Definition of terms

Coral specimen: Type of biological coral material collected, preserved, and processed (e.g., fragment, mucus, skeletal core).

Downstream analysis: The eventual laboratory analysis of the variable(s) of interest (e.g., chlorophyll concentration, lipid concentration, gene expression).

Fragment: A type of coral specimen in which a piece (e.g., branch/mound/plate) is removed from a colony growing on the reef. Includes skeleton, overlying tissue, and symbiotic/endosymbiotic microorganisms.

Genet1: Formed by sexual reproduction. All colonies and tissue that can trace their ancestry back to the same fertilization event belong to the same genet.

Parent colony: Coral colony growing on the reef from which specimens were removed. Of the studies which sampled multiple parent colonies, some specified that parent colonies were spatially separated by some minimum distance (e.g., 5 m) in order to minimize the likelihood of sampling genetically identical colonies. Other studies specified that genetic analyses were conducted in order to confirm that parent colonies were unique genets.

Sacrifice method: Any process which converts living coral tissue into non-living tissue.

Sampling design: The way in which specimens were sampled from a population such as details regarding the number of species which were sampled from the reef, the number of parent colonies sampled per species, and the number of specimens collected per parent colony.

Specimen collection: The removal of coral specimens from the reef or from experimental tanks.

Specimen preservation: The method by which coral specimens are sacrificed, preserved, and stored immediately following collection (e.g., snap-freeze with liquid nitrogen and stored at −80 ° C).

Specimen processing: Laboratory manipulation to prepare specimens for desired downstream analysis (e.g., airbrushing, freeze-drying, tissue homogenizing).

1Baums et al. (2019)

Survey Methodology

Literature search

This review was completed using methods modified from McLachlan et al. (2020). Briefly, a literature search of peer-reviewed publications was conducted using the ISI Web of Science database using the following search criteria: Title = coral, Topic = temperature AND bleach*. The original search returned 1,491 research publications between 1981 and 2020. This list was then refined to only include research articles published within the last 6.5-year period (January 1, 2014 to August 20, 2020). This temporal window was selected as our previous review revealed that almost half (46%) of coral bleaching research over the last 30 years has been published since 2014 (McLachlan et al., 2020). The remaining 689 publications were examined to assess if the study included the following elements: (1) at least one shallow-water (i.e., within the upper photic zone) coral species in the order Scleractinia, (2) the collection of coral specimens from the reef as part of a survey or for use in an experiment, and (3) preservation of specimens for downstream laboratory analyses, thus excluding studies which only conducted measurements on live coral fragments (e.g., photosynthesis, respiration, calcification, coral color). Studies that did not include those three elements were excluded. Studies using aquarium-cultured corals were also excluded as they did not have detailed information regarding where or how the corals were originally collected. One hundred and sixty-two studies met these criteria. Nine of these publications contained both a survey and an experimental component, or collected more than one specimen type (e.g., fragment and mucus). Each of these nine publications were treated as two independent studies (i.e., the survey vs. experimental portions, fragment vs. mucus study), bringing the total number of studies assessed to 171 (i.e., 162 + 9).

Table 1 Information collected and quantified in this review from coral bleaching surveys and bleaching experiments published between January 2014 and August 2020.

Information was split into six sections: (1) meta-data, (2) sampling design, (3) specimen collection, (4) specimen sacrifice, (5) specimen processing, and (6) downstream analyses.

(1) Meta-data	
1. Year of publication	
2. Category of study (bleaching survey or bleaching experiment)	
3. Author(s) and title of publication	
4. Journal of publication	
(2) Sampling design	
1. Coral family, genus, and species name	
2. Number of species sampled per study	
3. Number of parent colonies sampled per species	
4. Number of specimens sampled per parent colony	
5. Total number of specimens collected per study (determined from 2.2, 2.3, and 2.4 above)	
6. Total number of specimens collected per year (determined from 1.1 and 2.5 above)	
(3) Specimen collection	
1. Type of specimen collected from the reef (e.g., fragment, skeletal core, mucus, tissue)	
2. Tool(s) used to collect specimens (e.g., hammer and chisel, drill, syringe)	
3. If sterile techniquesa were used	
4. Sizeb of parent colony (cm or cm2)	
5. Sizeb of specimen (cm or cm2, applicable to fragments and skeletal cores only)	
6. Specimen transportationc methods	
7. Specimen transportationc duration (minutes)	
8. Type of specimen collected post-experiment (e.g., fragment, mucus, tissue)	
(4) Specimen preservation	
1. Method of specimen sacrifice (e.g., snap frozen with liquid nitrogen, ethanol preservation, airbrushed live)	
(5) Specimen processing	
1. Post-sacrifice processing techniques (e.g., airbrushing, grinding, homogenizing)	
2. Airbrushing methodsd	
3. Homogenization methodse	
4. Short-termf storage temperature (°C)	
(6) Downstream analysesg	
1. Number of downstream analyses conducted	
2. Type of downstream analyses conducted	
(A) Physiology (e.g., tissue biomass, chlorophyll concentration, lipid content)	
(B) -Omics (e.g., RNAseq, metagenomics, 16S amplicons, whole genome)	
(C) Microscopy and imaging (e.g., histology, electron microscopy, skeleton analysis)	
Notes.

a Includes sterilization of collection tools or use of sterile storage containers.

b In situations where authors reported a range of numerical values, the midpoint of the range was recorded. Example: “corals were 5–10 cm in length”, the midpoint range value is 7.5 cm. In several instances, authors did not specify which size metric (height vs. width/diameter) they were reporting (e.g., “we collected 20 fragments (5 cm)”). In these situations, we assumed values represented heights for branching morphologies, and diameters for massive/mounding morphologies.

c Transport refers to all steps taken between removal of sample from the reef and preservation (for survey type studies) or arrival at experimental location (for experimental studies).

d Airbrush or waterpik and the type of liquid involved (e.g., saltwater, freshwater, buffer).

e The equipment used in the homogenization of specimens, and the duration of homogenization in seconds.

f Any temperature at which the specimen was stored after sacrificing, during processing, and before the start of laboratory protocol for specific downstream analyses.

Data collection and analysis

For each study reviewed, the publication meta-data, details of the sampling design, and methods of specimen collection, preservation, and processing were recorded (Table 1). Meta-data included: year, category, authors, title, and journal of publication (Table 1.1). Sampling design information included: the name and number of species sampled, in addition to the number of parent colonies (Box 1) sampled, and number of replicates (i.e., number of specimens sampled per parent colony) (Table 1.2). This information was then used to calculate the total number of specimens collected per study and per year (Table 1.2). Detailed information about specimen collection from the reef or experimental tank was recorded, including any subsequent handling, storage, and transportation steps (Table 1.3). Specimen sacrifice (Box 1) and immediate preservation method, as well as any post-sacrifice specimen processing in preparation for downstream analyses was also recorded (Tables 1.3, 1.5). Twenty-two studies sacrificed and/or preserved specimens using two or more different methods. In these cases, all methods were recorded individually, thus increasing the sample size to 197 for specimen preservation and processing. Finally, the number and type of downstream analyses conducted were recorded in the following three categories: (1) physiology, (2) -omics, and (3) microscopy and imaging (Table 1.6). Note, the -omics category included amplicon sequencing and analyses of metagenomes, transcriptomes, proteomes, and metabolomes of corals and their symbiotic partners. All summary statistics and percentage data were calculated using Microsoft Excel (Microsoft Corporation, Redmond, WA, USA).

Results

Meta-data: almost two thirds of recent coral bleaching studies are manipulative experiments

Since 2014 the annual number of studies published that met our search criteria ranged from 14–30 (Table S1.1.1). In 2014, 2015, 2017, and 2018, more than two thirds of published manuscripts were experimental manipulation bleaching studies, while in 2016, 2019, and 2020, more than half of studies were surveys of bleaching in the field (Fig. S1). Of the 171 publications included in this review, 39% were surveys and 61% were manipulative experiments (Table S1.1.2). A total of 64 different scientific journals published coral bleaching-related research (Fig. S2, Table S2). However almost half of all studies were published in one of nine journals, the top two of which were Coral Reefs (12%) and Scientific Reports (9%) (Fig. S2).

Sampling design: at least 21,890 coral bleaching specimens have been collected since 2014

Specimens collected in bleaching surveys and bleaching experiments most commonly belonged to three coral families: Acroporidae (56%), Poritidae (36%), and Pocilloporidae 33% (Table S1.2.1). The top three species sampled were Acropora millepora (12%), Pocillopora damicornis (11%), and Stylophora pistillata (11%) (Fig. S3, Table S3). Eighty percent of studies sampled one or two species (Table S1.2.2). The median number of parent colonies sampled per species was 7, and the median number of specimens collected per parent colony sampled was 3 (Tables S1.2.3–1.2.4). Together, the median number of specimens collected per study (i.e., number of species × number of parent colonies per species × number of specimens per parent colony) was 60 (Table S1.2.5). Each year, between 2,000 to 5,000 specimens were collected (Fig. 2). Within the last 6.5 years, 21,890 coral specimens were collected in these bleaching studies (Fig. 2).

Specimen collection: coral fragments are the most common type of bleaching specimen

Almost 90% of coral bleaching specimens collected were fragments, 6% skeletal cores, 3% coral mucus, and less than 1% tissue or gametes (Fig. 3). Approximately one third of studies reported the tools used to remove specimens from the reef, of which the most commonly used were a hammer and chisel, drill, or bone cutters (Table S1.3.2). Five percent of studies specified that sterile techniques were used during specimen collection (Table S1.3.3), and the only collection tools described as sterile were syringes and cotton swabs. Although it was only reported in 8.2% of studies, the diameter of the parent colonies sampled ranged from 20 cm to 200 cm (Table S1.3.4). Sixty-three percent of studies reported the size of the coral specimen removed from the parent colonies, and of these, more than half reported the fragment height, a third reported fragment or skeletal core diameter, and a fifth reported the surface area (Table S1.3.5). The mean height of coral fragments collected was 4.25 ± 2.10 cm (mean ± 1SD) (Fig. S4). Of the 30% of studies that reported it, the duration of transport between specimen collection and specimen preservation ranged from 30 min to 12 h (Tables S1.3.6 and S1.3.7).

Figure 2 The total number of specimens collected during coral bleaching surveys and bleaching experiments published between January 2014 and August 2020 included in this review.

Full details in Table S1.2.6.

Figure 3 Percentage of 171 publications per type of specimens collected from the reef during coral bleaching surveys and bleaching experiments published between January 2014 and August 2020 included in this review.

Full data in Table S1.3.1.

Specimen preservation: five sacrifice methods are used by two thirds of studies

Twenty-one unique methods of specimen sacrifice were identified, and we consolidated them into three primary categories: (1) freezing, (2) chemical manipulation, or (3) mechanical tissue disruption (Table S1.4.1). Within the freezing category, the most commonly used method of specimen sacrifice was rapid ultra-freezing, sometimes referred to as snap freezing, with liquid nitrogen. Rapid ultra-freezing in liquid nitrogen was used by almost a quarter of studies, while 13% of studies sacrificed corals by placing them directly into −80 °C, −50 °C, or −20 °C freezer (Fig. 4). For sacrifice by chemical manipulation, a fifth of studies used a preservative and ∼7.5% of studies used a chemical fixative (Fig. 4). Five different chemical preservatives were described including ethanol, RNA stabilizing buffer, DNA stabilizing buffer, dimethyl sulfoxide (DMSO) buffer, and methanol (Table S1.4.1). Similarly, five different chemical fixatives were described including glutaraldehyde, formalin, paraformaldehyde, formaldehyde, and mercuric chloride (Table S1.4.1). Finally, for sacrifice by mechanical tissue disruption, 18% of studies airbrushed specimens directly following collection, while 4% ground the entire coral fragment using a mortar and pestle (Fig. 4). Overall, of the 21 unique methods of specimen sacrifice identified, almost two thirds of studies used one of five methods: (1) liquid nitrogen snap freezing, (2) −80 °C freezing, (3) ethanol preservative, (4) RNA buffer preservative, or (5) airbrushed live (Table S1.4.1).

Figure 4 Percentage of 171 publications using each of the specimen sacrificing methods in coral bleaching surveys and bleaching experiments published between January 2014 and August 2020 included in this review.

Specimen processing: a wide variety of airbrushing and homogenizing methods identified

Various processing techniques were used to alter the specimen state, following sacrifice, to prepare it for downstream analyses. Post-sacrifice processing techniques included: airbrushing, homogenizing, freeze-drying, oven or air drying, and the addition of other chemicals or preservatives. Forty percent of studies used an airbrush or waterpik to separate coral tissue from the coral skeleton (Table S1.5.2). Of these, six different airbrushing methods were described, the most common of which was airbrushing with saltwater, followed by airbrushing with a buffer (Table S1.5.2). Seventy-six studies (39%) homogenized tissues –a fifth of which did so with an electric homogenizer (e.g., TissueTearer™) and another fifth used a mortar and pestle, although several other methods were also described (Table S1.5.3). The short-term storage temperatures used during specimen processing was reported in 59% of studies, and the most frequently used storage temperature, used by a quarter of studies, was −80 °C (Table S1.5.4).

Downstream analyses: symbiodiniaceae density, identification, and chlorophyll concentration are the most common

Following specimen preservation and processing, we identified 29 different downstream analyses that were conducted (Table S1.6.1). We categorized the analyses into three broad categories: (1) physiology, (2) -omics, and (3) microscopy and imaging (Fig. 5). The most frequently measured physiological variables were chlorophyll concentration (28%) and total soluble protein (18%) (Fig. 5A). In the -omics category, 29% of studies taxonomically identified Symbiodiniaceae species, 13% performed transcriptomic and gene expression analyses, and 11% analyzed one or more aspects of the coral microbiome (Fig. 5B). Within the microscopy and imaging category, the top two downstream analyses were Symbiodiniaceae density quantification (39%) and coral skeleton X-ray imaging (5%) (Fig. 5C). Forty-two percent of studies measured only one downstream analysis (Table S1.6.2). The proportion of studies which conducted downstream analyses from one, two, or three of the categories (i.e., physiology, -omics, and microscopy and imaging) is summarized using a Venn diagram (Fig. 6). Thirteen percent of studies conducted at least one downstream analysis from each of the three categories (i.e., center of the Venn diagram, Fig. 6).

Figure 5 Downstream analyses conducted in coral bleaching surveys and bleaching experiments published between January 2014 and August 2020 included in this review.

Percentage of 197 publications which conducted downstream analyses in the categories (A) physiology, (B) -omics, and (C) microscopy and imaging HPLC = High-performance liquid chromatography; DMSP/DMSO = dimethyl sulfoniopropionate/dimethyl sulfoxide; MS = mass spectrometry. Additional details in Table S1.6.1.

Figure 6 Overlap in downstream analyses conducted in coral bleaching surveys and bleaching experiments published between January 2014 and August 2020 and included in this review.

Illustrated in the Venn diagram above are the percentage of studies that conducted at least one downstream analysis within each category. For example, 6.6% of studies conducted at least one physiological and one -omic analysis, but none in the category microscopy and imaging.

Half of bleaching studies handle specimens using six methodological pathways

From these 171 studies, data synthesis regarding specimen collection, preservation, and processing revealed 37 distinct methodological pathways prior to downstream analyses (Fig. 7). However, while these methods were varied, six methods were found to encompass almost half of all bleaching studies. These top six pathways for coral specimen manipulations were as follows: (1) sacrifice via snap freezing with liquid nitrogen followed by airbrushing, (2) sacrifice via airbrushing followed homogenization and freezing in conventional (i.e., not ultrarapid/blast) freezer, (3) sacrifice via freezing in conventional freezer followed by airbrushing, (4) sacrifice by ethanol preservation with no further processing, (5) sacrifice via snap freezing with liquid nitrogen followed by grinding, and (6) sacrifice via chemical fixative followed by decalcification (Fig. 7).

Figure 7 Heat map summarizing methodological pathways and downstream analyses categorized in this review.

Cells shaded based on number of studies ranging from 1 (pale) to >9 (dark). AB, airbrush; BL, bleach (NaOCl); CRYO, cryopreserved; DMSO, Dimethyl sulfoxide; DMSP/DMSO, dimethyl sulfonio-propionate /dimethyl sulfoxide; DNA, DNA buffer; DNS, did not state; ETH, ethanol; FIX, fixative; FR, frozen with conventional freezer; G, gametes. GLY, glycerol; GR, ground; HPLC, High-performance liquid chromatography; METH, methanol; RNA, RNA buffer; SNAP, snap frozen with liquid nitrogen. The underlying raw data used to construct this table is in the Supplement 2.

Conclusions

Thousands of invaluable coral bleaching specimens were collected over the last 6.5 years

In this study, we reviewed 66 coral bleaching surveys and 105 coral bleaching experimental studies published between January 2014 and August 2020. Our data indicate that 21,890 coral specimens (Fig. 2), primarily fragments (Fig. 3), were collected from 122 scleractinian species over the last 6.5 years. Using the mean size of fragments collected (∼4 cm height and ∼6 cm diameter, Fig. S4, Table S1.3.5), we calculated the approximate volume of each fragment collected to be 113 cm3, assuming each fragment was cylindrical in shape (i.e., mean volume of cylinder = π r2 ×h = 3.14 × 32 ×4 = 113 cm3). Thus, the total volume of coral fragments collected since January 2014 is ∼2,473,570 cm3 or 2.47 m3 (i.e., 21,890 specimens x 113 cm3), which is equivalent to five upright −80 °C freezers full of coral fragments (assuming a freezer capacity of 490 L). While the total number of fragments collected may seem large, the total volume of coral material removed from reefs globally over a 6.5-year period appears negligible. As the publications included in this review represent only a subsample of all bleaching research published since 2014 (i.e., only those which matched our search criteria), the number of specimens collected is almost certainly an underestimate. These specimens are highly valuable in terms of the biological and climatological information they possess, as well as the financial, temporal, and labor-costs associated with their procurement. For example, manipulative experiments are excellent tools for investigating mechanistic processes underlying coral bleaching as they can control various aspects of the environment (e.g., water movement, light, depth) and underlying biology (e.g., genotype), thus increasing the signal-to-noise ratio in the resulting data. However, manipulative experiments are also expensive, labor-intensive to set up and maintain, and require a durational commitment to observe the potential effects of explanatory variables ranging from hours to months (McLachlan et al., 2020). Conversely, observational surveys have the advantage of cataloguing natural variation in corals (e.g., in situ responses of corals during bleaching events) by encompassing the full range of abiotic and biotic factors which cannot be recreated experimentally. Therefore, coral archived following both experimental manipulation and observational surveys are precious and important scientific specimens. Given that coral bleaching specimens are both expensive to collect and likely to become a limited resource in future, maximizing the utility of such specimens for additional analytic methods can expand the overall investigative yield of each experiment or survey.

Similarities in methodologies indicate a large potential for future collaborations

In order to maximize the scientific output from coral bleaching specimens, it is essential to consider the methodologies used during sampling, sacrifice, and processing as many downstream analyses are variably sensitive to upstream handling and processing. Our review revealed several commonalities and some differences in the way in which coral bleaching specimens are handled following removal from the reef. For instance, we found that specimen ‘transport duration’ had a large range from 30 min to 12 h (Table S1.3.7). Given that the amount of time between specimen collection and freezing (or “freezing delay”) is an important variable that influences specimen integrity for some analyses, this initial step may be a critical target to amend in future studies that aim to conduct more downstream analyzes. In cancer research, for example, a freezing delay of 30 min or more significantly alters both the metabolomic (Haukaas et al., 2016) and gene expression profiles (Cecco et al., 2009) of tumor specimens. In corals, a study investigating black band disease found that specimens frozen immediately after collection contained more proteobacterial 16S rRNA sequences with more cyanobacterial and sulfur-oxidizing bacterial sequences compared to unfrozen specimens analyzed within 1 h of collection (Sekar, Kaczmarsky & Richardson, 2009).

In terms of specimen preservation, 21 different methods of sacrifice were identified, suggesting that there is large variation in preservation methodologies (Table S1.4.1). However, over 75% of bleaching studies sacrificed their specimens using just five of these methods. This common framework in preservation practices can increase the potential for sharing of specimens among laboratories that use comparable methods. When researchers do share specimens and/or conduct additional downstream analyses, it is essential to consider how the methods of specimen collection and sacrifice could potentially influence the results. For example, it has been shown that the use of fixatives such as formalin and mercuric chloride in specimen preservation cause significant increases in the δ15N and δ13C values of the muscle tissue of flounder fish compared to preservation by freezing (Bosley & Wainright, 1999). In corals, a study by Hernandez-Agreda, Leggat & Ainsworth (2018) found significant changes in the relative abundance and dominance of certain coral microbial taxa due to differences in specimen preservation (i.e., snap freezing vs. DMSO preservation vs. paraformaldehyde fixation). The method of specimen preservation is also important to consider. We found that almost half of all studies airbrushed coral fragments (Table S1.5.2), and 38% homogenized tissues (Table S1.5.3) indicating consistency in the way in which specimens are being processed. However, within each of these two processing methods, we catalogued variety in the airbrushing mediums used (e.g., 44% saltwater vs. 18% buffer, Table S1.5.2) and the type of homogenizing tools (e.g., 21% electric vs. 20% mortar and pestle, Table S1.5.3). A comparative study by Pupier, Bednarz & Ferrier-Pagès (2018) found that specimen state (i.e., frozen or freeze-dried) and homogenization media (i.e., saltwater vs. freshwater) significantly affected the quantification of some tissue parameters in soft corals such as chlorophyll pigments and protein concentration. A study by Krediet et al. (2015) compared methods of quantifying Symbiodiniaceae cell density in anemones and found differences in Symbiodiniaceae abundances depending on whether specimens were (a) frozen whole, thawed, and homogenized compared to those (b) homogenized, frozen, and then thawed. Conlan, Rocker & Francis (2017) compared the use of airbrushed versus ground coral specimens, and found that total lipid content and the relative proportion of triacylglycerols and fatty acid molecules was significantly underestimated in airbrushed specimens due to the retention of organic material within the skeletal organic matrix. Finally, the method of homogenization (i.e., bead homogenization vs. grinding) was shown to influence investigations of the coral microbiome (Hernandez-Agreda, Leggat & Ainsworth, 2018). These studies illustrate that careful consideration of specimen collection, preservation, and processing is required in order to accurately compare among studies.

Overall, despite identifying 37 distinct methodological pathways—from collection to processing of specimens—our synthesized data revealed that almost 50% of all bleaching studies used one of six methodological pathways (Fig. 7). These common frameworks in collection, preservation, and processing methods illustrate that archived coral specimens could be readily shared among researchers for additional analyses. Our detailed dataset of the 177 studies (see Table S4) documents which downstream analyses have already been conducted on potentially archived specimens and could be used as a tool for researchers/collaborators to identify specimens that may be available for additional analytical investigations. However, it was not possible to ascertain from the published literature what proportion of 21,890 collected specimens were archived and potentially available for additional measurements. Further, many methods consume or destroy most or all of the specimen material during their analytical steps, thus rendering them unavailable for further analyses. Regardless, even if only a quarter of these specimens are archived, then there should be almost 5,500 coral bleaching specimens potentially available for further analytical investigation through future collaborations. In addition, this review provides a methodological roadmap for researchers to refer to when considering future survey or experimental work if they would like to optimize the number of possible downstream analyses and maximize the utility of new specimens within the context of their study.

Complementary methods to bridge the gap between physiology, -omics, and microscopy and imaging

Despite asking widely different questions about the causes and mechanisms of coral bleaching, we found that there are a remarkable number of common methodologies used across the 3 primary disciplines we identified in this study: physiology, -omics, and microscopy and imaging (Fig. 5). For instance, the most frequently used collection and preservation pathway in all studies was to ultra-rapidly freeze specimens using liquid nitrogen followed by airbrushing the coral tissues (Fig. 7). Our heat map shows that 13 different downstream analyses can be conducted following this 2-step pathway including: chlorophyll concentration and HPLC pigment analyses, total soluble protein and carbohydrate, ash-free dry weight tissue biomass, Symbiodiniaceae identification, microbiome, enzymatic assays, metabolomics, Symbiodiniaceae density and mitotic index (Fig. 7). However, the mean number of downstream analyses conducted per publication reviewed here was only two (Table S1.6.2). This discrepancy suggests that there is significant opportunity to further maximize the number of analyses conducted per specimen, and how coral specimens are used and shared across discipline-specific analyses. For example, we found that over a third of studies conducted omics-based analyses only. However, the way specimens are collected for -omics analyses (e.g., snap freezing, preserved in ethanol, or preserved in RNA buffer) would be suitable for a range of additional downstream analyses in both the physiology and microscopy and imaging categories as well (Fig. 7).

While in theory, the sharing of specimens with similar collection and preservation methods is simple, we recognize that there are several practical factors which may limit the capacity for a research group to archive specimens for additional future analyses, such as funding, time limitations, supplies and resource availability, freezer space, opportunity costs, and expertise. Nevertheless, there are likely several low-cost changes that can be made during specimen collection, sacrifice, preservation, processing, and long-term archiving which would facilitate the sharing of specimens collaboratively in the future to optimize the total number of possible downstream analyses for every specimen. For instance, if the primary goal of a study is to collect coral specimens for Symbiodiniaceae density enumeration, but the researchers were able to use sterile handling and collection techniques, then specimens collected would also be suitable for downstream analysis of the microbiome. A recent seminal paper by Greene et al. (2020) outlined an optimized and standardized protocol for collecting coral specimens such that researchers could maximize their potential and be used for microbial, metabolomic, and histological analyses simultaneously. Additional highly detailed papers such as this are needed to further guide future researchers along methodological pathways that maximize the utility and comparability of coral specimens for multiple downstream analyses.

Reporting of methodological information and future directions

During data collection for this review, we found several instances where some details concerning specimen handling were not provided (Fig. 6). Of the five types of methodological information reviewed (Fig. 1), detailed information regarding how corals were collected was most frequently omitted (Fig. 8B). For instance, 95% of studies did not report the size of the parent colony from which they sampled (Fig. 6A). This information could be important for many data interpretations because colony size may be associated with colony age (assuming no asexual fragmentation has occurred (Hughes, 1984)). A specimen taken from a young colony (e.g., <5 cm diameter depending on the species) may respond differently to in situ bleaching or experimental treatments than a specimen taken from a larger conspecific (e.g., >5 cm, Álvarez Noriega et al., 2018). Likewise, colony size is a major factor determining some coral characteristics, such as microbiome richness (Pollock et al., 2018). Details regarding the method and duration of specimen transport following collection were also not reported in 70% and 95% of studies, respectively (Fig. 8B). Information on the method of sacrifice and short-term storage temperature was missing from 11% and 41% of studies, respectively (Figs. 8C, 8D). Yet, details regarding the handling, sacrifice, and preservation of specimens are critical for effective sharing of this material with other researchers. For example, 95% of studies did not state whether sterile tools were used at any stage of the methodology (Fig. 8B). We assume that not reporting the use of sterile equipment is likely because non-sterile equipment was used, but it would be advantageous when proposing specimen sharing and collaboration if this information were explicitly stated in future manuscripts. The adoption of more sterile techniques (e.g., wearing gloves or using sterile vessels for storage) during specimen collection is an example of a small low-cost methodological workflow alteration that could substantially maximize research output per specimen and facilitate potential collaborations for microbiome investigations, resulting in a more holistic view of coral bleaching response.

Figure 8 Methodological reporting information for coral bleaching surveys and bleaching experiments published between January 2014 and August 2020 included in this review.

Percentage of studies that reported (light blue bars) or did not report (dark blue bars) methodological information relating to (A) sampling design, (B) specimen collection, (C) specimen preservation, and (D) specimen processing. Grey bars show publications for which data was not applicable (e.g., studies which only collected mucus would not report the size of fragment collected). Likewise, not all studies conducted airbrushing or homogenizing during specimen processing.

Overall, we recognize that there are several factors which may account for the missing information in the coral bleaching studies reviewed here, including restrictive word limits in scientific journals or simply that the data was not pertinent to the aims of the study (e.g., use of sterile techniques). One solution to this would be to construct an outlet which allows researchers to retroactively provide missing/additional data about their studies and to encourage comprehensive metadata reporting in supplemental materials of publications. To enhance the potential for sharing samples and leveraging additional research from current and future studies, a common framework was developed as part of the second Coral Bleaching Research Coordination Network (CBRCN) 2020 and is presented in our companion paper (Vega Thurber et al., unpublished data). Moving forward, it would also be helpful to convert the dataset from this study into an online, interactive platform. Together, the common framework and searchable database could help to accelerate the rate of scientific discovery in coral bleaching research in the face of a warming climate.

Supplemental Information

Supplemental Information 1 Number of coral bleaching surveys (light blue) and bleaching experiments (dark blue) published between January 2014 and August 2020 included in this review

Additional data in Table S1.1.1

Click here for additional data file.

Supplemental Information 2 Percentage of publications in each scientific journal where more than 1% of coral bleaching surveys and bleaching experiments included in this review were published between January 2014 and August 2020

See Table S2 for complete list.

Click here for additional data file.

Supplemental Information 3 Coral species included in more than 2% of coral bleaching surveys and bleaching experiments published between January 2014 and August 2020 included in this review

See Table S3 for complete list.

Click here for additional data file.

Supplemental Information 4 Boxplot of coral specimen height, diameter, or surface area as reported in coral bleaching surveys and bleaching experiments published between Janurary 2014 and August 2020 included in this review

Sizes were applicable only to specimens which were fragments or skeletal cores, thus excluding mucus, tissue, and gamete specimens. Details in Table S1.3.5.

Click here for additional data file.

Supplemental Information 5 Summary statistics and percentage data for coral bleaching surveys and bleaching experiments conducted between January 2014 and August 2020 included in this review

Data categories include: (1) meta-data, (2) sampling design, (3) specimen collection, (4) specimen preservation, (5) specimen processing, and (6) downstream analyses. The number of studies in each category is in parentheses. The total number of studies in each category (second column) were used as the denominator in calculating the percentages.

Click here for additional data file.

Supplemental Information 6 Scientific journals in which coral bleaching surveys and bleaching experiments were published between January 2014 and August 2020 included in this review

Click here for additional data file.

Supplemental Information 7 Scleractinian coral species collected during coral bleaching surveys and bleaching experiments in studies published between January 2014 and August 2020 included in this review

Details regarding exactly which studies (i.e., author, year, and study title) used which species can be found in Supplement 2.

Click here for additional data file.

Supplemental Information 8 Raw data of specimen collection, preservation, and processing methods used during coral bleaching surveys and bleaching experiments in studies published between January 2014 and August 2020 included in this review

Click here for additional data file.

The authors would like to thank the members of the Coral Bleaching Research Coordination Network (CBRCN) for their feedback following the first presentation of the preliminary data for this review during the second CBRCN Workshop: Sample Preparation and Archiving via Zoom (30 June–4 July 2020).

Additional Information and Declarations

Competing Interests

Author Contributions

Data Availability

The authors declare there are no competing interests.

Rowan H. McLachlan performed the experiments, analyzed the data, prepared figures and/or tables, authored or reviewed drafts of the paper, and approved the final draft.

Kerri L. Dobson and Emily R. Schmeltzer performed the experiments, authored or reviewed drafts of the paper, and approved the final draft.

Rebecca Vega Thurber and Andréa G. Grottoli conceived and designed the experiments, authored or reviewed drafts of the paper, and approved the final draft.

The following information was supplied regarding data availability:

The raw data of all methodological information are available in the Supplemental Files.

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
