# Peer review of "A review of coral bleaching specimen collection, preservation, and laboratory processing methods"

_PeerJ, doi:10.7717/peerj.11763_

## Round 0.1 · original submission · Minor Revisions

The paper will be accepted after correcting those minor edits.
Great work.

·

Basic reporting

This manuscript provides simple and elegant insight into how methodology varies in the field of coral bleaching research. I have limited criticism, mostly consisting of grammatical or typographical errors. The introduction is adequate, the justification for the study clearly identified.

Experimental design

The main point thatI think needs addressing is why the limited time frame was chosen, as the full dataset from 1981 surely would provide interesting temporal insight. Further, I recognise that this manuscript was likely developed at the end of 2020 and for that reason only a partial year is included, however since we are now in May 2021 it would be best if the authors could complete this final year’s dataset to increase comparability between years. On that note, it may be interesting to comment briefly on the temporal patterns of sample collection – i.e. is there evidence for increased sample collection coinciding with/following bleaching events?

Validity of the findings

Conclusions are well stated and well supported.

Additional comments

- Line 153 – why was this particular 6.5 year window selected? Some justification is needed.
- Line 155 – how did you define “shallow-water” species?
- Line 176 – “included” should be “including”
- Line 195 – I believe this should be Fig S1
- Line 303 – “out” should be “our”
- Line 341 – “who” should be “that”
- Line 344 – “causes” should be “cause”
- Line 357 – what specifically do you mean by tissue parameters?
- Line 424 – I don’t think the snippet in brackets is necessary
- Line 430 – missing second closed bracket after reference
- Lines 433-434- colony size can even influence bleaching susceptibility https://link.springer.com/article/10.1007%2Fs00338-018-1677-y
- This may be a personal preference, but references to figures / tables in the discussion is distracting to me. Perhaps these can be reduced.

·

Basic reporting

no comment

Experimental design

no comment

Validity of the findings

no comment

Additional comments

In ‘a review of coral bleaching specimen collection, preservation, and laboratory processing methods’ McLachlan review and discuss the common practices for coral collection and processing in coral bleaching research over the last 6.5 years. In general, the manuscript is well written and will appeal to the broad readership of Peer J. The authors have done well to synthesize the large volume of research in this field and present a nice and clear manuscript, well done. However, I feel what was ultimately lacking was a ‘best practices’ section to direct future research. Based on what we know about sample degradation (from other models if not corals) etc, what is the best approach for preservation and processing for multiple downstream analyses? Obviously, this depends on the questions and methods, but perhaps a diagram or flow chart that could walk researchers through best practices depending on the desired data? I think such a section would be extremely valuable to other researchers when planning field work and research.

---

## Round 0.2 · accepted · Accept

The manuscript provides simple and elegant insight into how methodology varies in the field of coral bleaching research, which will be helpful for many coral reef scientists.